# Determinants of cognitive impairment in multiple system atrophy: Clinical and genetic study

Amina Nasri[1,2,3], Alya Gharbi[1,2,3☉], Ikram Sghaier[1,3☉], Saloua Mrabet[1,2,3], Amira Souissi[1,2,3], Amina Gargouri[1,2,3], Mouna Ben Djebara[1,2,3], Imen Kacem[1,2,3], Riadh Gouider[1,2,3]*

**1** Neurology Department, LR18SP03, Razi Universitary Hospital, Tunis, Tunisia, **2** Faculty of Medicine of Tunis, University of Tunis El Manar, Tunis, Tunisia, **3** Clinical Investigation Center (CIC) "Neurosciences and Mental Health", Razi Universitary Hospital, Tunis, Tunisia

☉ These authors contributed equally to this work.
* riadh.gouider@gnet.tn

**Data Availability Statement:** All relevant data are within the paper and its Supporting information files.

## Abstract

### Introduction

Classically, cognitive impairment (CI) was not considered as a paramount feature of multiple system atrophy(MSA) in both parkinsonian(MSA-P) and cerebellar(MSA-C) motor-sub-types. Yet, growing evidence indicates currently the commonness of such deficits among MSA patients in different populations. Our aim was to evaluate the cognitive profile of MSA Tunisian patients and to analyze the underlying clinical and genetic determinants

### Methods

In a retrospective cross-sectional study, clinically-diagnosed MSA patients were included. All subjects underwent clinical and neuropsychological assessments to characterize their cognitive profile. The associations with their *APOE* genotype status were analyzed. Determinant of CI were specified.

### Results

We included 71 MSA patients. Female gender(sex-ratio = 0.65) and MSA-P subtype(73%) were predominant. Mean age of disease onset was 59.1years. CI was found in 85.7% of patients(dementia in 12.7% and Mild cognitive impairment(MCI) in 73% of patients mainly of multiple-domain amnestic type(37.3%)). Mean MMSE score was lower among MSA-P compared to MSA-C(23.52 *vs.* 26.47;p = 0.027). Higher postural instability gait disorder(PIGD) and MDS-UPDRS-III scores were noted in demented MSA patients(p = 0.019;p = 0.015 respectively). The main altered cognitive domain was attention(64.8%). Executive functions and mood disorders were more affected in MSA-P(p = 0.029,p = 0.035 respectively). Clinical and neurophysiological study of dysautonomia revealed no differences across cognitive subtypes. *APOE* genotyping was performed in 51 MSA patients with available blood samples. Those carrying *APOEε4* had 1.32 fold higher risk to develop CI, with lower MMSE

**Funding:** The authors received no specific funding for this work.

**Competing interests:** The authors have declared that no competing interests exist.

score(p = 0.0001). Attention and language were significantly altered by adjusting the p value to *APOEε4* carriers(p = 0.046 and p = 0.044 respectively). Executive dysfunction was more pronounced among MSA-P *APOEε4* carriers(p = 0.010).

## Conclusion

In this study, the main determinants of CI in Tunisian MSA patients were MSA-P motor-subtype, mainly of PIGD-phenotype, disease duration and *APOEε4* carrying status, defining a more altered cognitive phenotype. This effect mainly concerned executive, attention and language functions, all found to be more impaired in *APOEε4* carriers with variable degrees across MSA motor-subtypes.

## Introduction

Multiple system atrophy (MSA) is a rare adult-onset and fatal neurodegenerative disorder, characterized by progressive autonomic and motor dysfunction. The pathological hallmark of MSA is the emergence of glial cytoplasmic inclusions (GCIs) corresponding to insoluble α-synuclein accumulations within oligodendrocytes, found particularly in the striatonigral and olivopontocerebellar systems in the brain [1]. Nowadays, it is established that MSA belongs to the group of synucleinopathies alongside with Lewy body disorders [1, 2]. The pathological distribution of lesions in MSA brains leads to multiple phenotypes. According to the predominant clinical features, MSA may be classified into parkinsonian (MSA-P) and cerebellar (MSA-C) motor subtypes [3, 4].

Recently, non-motor and non-autonomic signs have been increasingly recognized in MSA. Dementia or major neurocognitive impairment was considered as exclusion criteria in the first consensus diagnostic criteria [3]. Previously, it has been admitted that cognition remained preserved in MSA. However, cognitive impairment (CI) was reported in up to 37% of neuro-pathologically proven MSA cases [5]. Lately, this CI was more frequently described in MSA. The most common presentation is mild neurocognitive impairment (MCI) with frontal and executive dysfunction [6, 7]. Nonetheless, the cognitive spectrum in MSA patients seems to be broader, with cognitive decline affecting in other domains [8–10]. The latter could lead to major neurocognitive impairment that interferes with daily living activities [11]. Cognitive deficits can also be found in early stages of the disease in up to 22% of patients [12] and in some patients as a prodromal premotor symptom [13]. Recent follow-up studies showed that cognitive functions may have different evolution over disease course in the two motor subtypes of the disease [14]. Three different MSA cognitive profiles were described upon follow-up comprising normal cognition, stable selective attention-executive deficits, and progressive attention–executive dysfunction associated with memory and visuospatial impairment evolving over time into dementia [15].

Data from previous studies remain though inconsistent about the neuropsychological and clinical heterogeneity of cognitive profile in MSA [16] and the predictive factors of CI in this disease [17]. Among these possible factors, the role of autonomic dysfunction, assessed with various tools including neurophysiological tests, has been previously suggested [5, 18]. In addition to demographic and clinical factors, the implication of genetic factors in cognition remains understudied in MSA. In fact, although the genetic architecture of MSA is still shaping, genotype-phenotype and pathological correlations are hardly depicted [6, 9, 7, 19]. The Apolipoprotein E (*APOE*) gene which is known to be strongly associated with cognitive decline

in neurodegenerative diseases, could be a possible candidate. Indeed, *APOE* has three common alleles; ε4, ε3, ε2 [8, 10, 20]. *APOE* ε4 is the major genetic risk factor of late onset Alzheimer's disease (AD) and has been associated with cognition in other synucleinopathies like dementia with Lewy bodies (DLB) [21, 22]. Although *APOE* gene has been seldom assessed in MSA [6], its link with CI in MSA patients was even less explored. The aim of this study was to evaluate the cognitive profile of MSA patients and to analyze clinical and genetic determinants of CI in MSA.

## Material and methods

### 1. Study subjects

A cross-sectional retrospective study was carried in the Department of Neurology at Razi University Hospital, a tertiary care hospital in Tunis in North Tunisia, over a period of 18 years (from January 2003 to December 2020). Patients with clinically diagnosed probable or possible MSA according to the revised Gilman Criteria were included [3]. All patients had a neurological examination performed by a movement disorder specialist and a systematic brain imaging. We excluded all patients with parkinsonism and/or ataxia of other origin.

### 2. Clinical and neuropsychological assessment

Detailed neurological examination was performed in all patients to collect anamnestic data and to evaluate the severity of motor and non motor impairment. Information was obtained from the participants and their caregivers about family and personal medical history and medication. Demographic, clinical and neuropsychological data were collected using standardized case-report forms. Disease onset was defined as the age of occurrence of either motor symptoms (parkinsonism and/or cerebellar ataxia) or autonomic dysfunction. Diagnostic delay was defined as the difference between the age at first consultation and the age of disease onset. We defined the type of initial symptoms and specified the age at onset of motor, autonomic, subjective cognitive and psychiatric symptoms (memory, language, hallucinations, depression, and behavioral disorders). For cognitive symptoms, we determined their impact on daily living by the Katz Index of Independence in Activities of Daily Living (ADL) and the Lawton Instrumental Activities of Daily Living scale (IADL) scales to assess functional capacity [23]. Autonomic symptoms included; lower urinary tract symptoms (LUTS), erectile dysfunction, orthostatic hypotension, constipation, and sudomotor symptoms. Stridor and sleep disorders, mainly RBD (rapid eye movement sleep (REM) behavior disorders) were specified. MDS-Unified Parkinson's disease Rating Scale section III (MDS-UPDRS-III) was used to rate the severity of extrapyramidal symptoms. Parkinsonian motor phenotypes were classified into three subgroups, tremor dominant (TD), postural instability gait disorder (PIGD) or indeterminate phenotypes, according to the published formulas used in Parkinson's disease (PD) [24]. The degree of severity of parkinsonian symptoms was measured using the Hoehn &Yahr (H&Y) scale. For the assessment of levodopa responsiveness, we used an acute pharmacological test, namely the acute levodopa challenge (ALC), which is routinely performed in our department. We used a standard protocol by administering a single dose of levodopa/ carbidopa 250/25 mg. Motor response was quantified using the MDS-UPDRS-III. During the ALC, motor examination was performed immediately before and every 30 minutes after levodopa intake until the motor conditions returned to the motor baseline status. We calculated the percentage of motor response as the ratio of the difference between the baseline and the peak-of-dose motor scores by the baseline motor score.

$$\frac{\text{Baseline motor score} - \text{the peak of dose motor score}}{\text{Baseline motor score}} \times 100 = \%$$

Levodopa-responsiveness was defined as an improvement rate ≥30% of MDS- UPDRS-III [25].

Other motor signs were assessed on examination including other movement disorders, cerebellar, pyramidal, bulbar and oculomotor signs. Patients were classified into two motor subtypes: MSA-P and MSA-C. They were also classified into probable or possible MSA according to the revised 2008 Gilman Criteria, specifying the dysautonomic features that allowed this probability subtyping (urogenital dysfunction and/or orthostatic hypotension) [3].

Each patient underwent a neuropsychological examination at first consultation comprising the 30-item mini-mental state examination (MMSE) to assess overall cognitive efficiency. The MMSE has been standardized and validated in Tunisia and adjusted for age and education [26]. The frontal assessment battery (FAB) was used to evaluate executive functions and a score less than 16 was considered abnormal [27]. Episodic memory was assessed with the Free and Cued Selective Reminding Test (FCSRT) of Grober & Buschke. Visuo-spatial functions were evaluated by the Clock-Drawing Test [28]. Beck's Depression Inventory (BDI) (if the age <65 years) [29] and Geriatric Depression Scale (GDS) (if the age≥65 years) were used to evaluate mood disturbance and detect depression [30]. Behavioral disorders were identified by the Neuropsychiatric Inventory (NPI) [31]. Other domains evaluated by neuropsychological assessment included orientation, attention, apraxia, agnosia, judgment and reasoning. Language was evaluated with a speech assessment battery including verbal fluency, grammar and syntax, comprehension, repetition, articulation, semantic knowledge, reading and writing evaluation. We classified the patients cognitively in three subgroups: 1-No cognitive impairment (NCI) (in the absence of subjective or objective cognitive deficits); 2-MCI and 3-dementia or major neurocognitive disorder according to the Diagnostic and Statistical Manual of Mental Disorders (DSM-5) [32]. MCI patients were further subdivided in four clinical subtypes: 1-single-domain amnestic MCI; 2- multiple-domain amnestic MCI; 3- single-domain non amnestic MCI; and 4- multiple-domain non amnestic MCI [32].

## 3. Neurophysiological autonomic testing

Parasympathetic cardiac control was assessed by three tests: heart rate variation to deep breathing (HR-DB), to Valsalva maneuver (HR-V) and to standing (HR-S). Patients were graded as per Ewing's criteria for parasympathetic dysautonomia based on Ewing's heart rate tests battery into normal, early or definite autonomic dysfunction. The sympathetic autonomic system was assessed with the SSR evaluated at the four limbs. Absent SSR at either one or more extremities was regarded as abnormal independently from its latency and/or amplitude. The absence of SSR in at least one limb defined sympathetic autonomic dysfunction [33].

## 4. Genetic study

Genotyping of *APOE* was performed using the Restriction Fragment Length Polymorphism Polymerase Chain Reaction. *APOE* genotypes were determined by scoring for a unique combination of fragment sizes, as depicted by Hixon et al. In fact, digestion by HhaI restriction enzyme gives various combinations of fragment sizes for each genotype as pursue: ε2/ ε2, 91 and 83 bp; ε3/ ε3, 91 and 48 bp; ε4/ ε4, 72 and 48 bp and a mixed genotype: ε2/ ε3, 91, 83, and 48 bp; ε3/ ε4, 91, 72 and 48 bp; ε2/ ε4, 91, 83, 72 and 48 bp.

## 5. Statistical analysis

We described demographic and clinical characteristics as well as *APOE* genetic frequencies then assessed the relationship between the clinical variables and *APOE* genotype in the total cohort and compared them across subgroups. Continuous variables were expressed as mean±

standard deviation, while the median and the 1$^{st}$ quartile and 3$^{rd}$ quartile were used to measure
the central tendency in the case of skew distribution with some extreme values. Differences in
the proportions were analyzed by the Chi-square test and the Fischer's exact test. Multinomial
logistic regression was used to model outcome variables according to *APOE* ε4 carrying status.
A value of $p < 0.05$ was considered statistically significant. Spearman rank correlation test was
used to evaluate correlations. Corrections for multiple comparisons were employed with a
Bonferroni correction. All statistical procedures were performed with R software for Windows
using the "SNPassoc","multinom", "Hmisc" and "ggplot2" packages.

## 5. Ethics

All subject investigations conformed to the principles outlined in the Declaration of Helsinki
and have been performed with permission of the Razi hospital ethic committee. All subjects
were informed about the purposes of the study and gave a written consent (patients themselves
or caregivers) to participate in the study.

## Results

### 1. Clinical and paraclinical findings in MSA patients

We included 71 patients diagnosed with MSA (mainly of MSA-P motor subtype (73.23% vs.
26.76% of MSA-C). The diagnosis of MSA was probable in 73% of cases (65% with urinary
incontinence and erectile dysfunction in males, 7% with both rigorously defined urogenital
symptoms and orthostatic hypotension and only one patient with orthostatic hypotension)
and possible in 27%of patients. The mean follow-up of the total cohort was 3.45 years [ranging
from 1 year to 11 years]. Detailed characteristics of total MSA patients as well as stratified
according to motor subtype (MSA-P and MSA-C) were summarized in Table 1.

Female predominance was noted with sex-ratio equal to 0.65. Mean age of disease onset
was 59.11±9.01years with no significant differences between MSA motor subtypes (p = 0.495).
The median age of parkinsonism onset among the total group was 62 years ranging from 37 to
77. Parkinsonism duration was significantly shorter in MSA-C (p = 0.0051).

Clinically, all patients reported autonomic dysfunction. Among autonomic symptoms,
LUTS dysfunction was the most prevalent (85.91%) followed by orthostatic hypotension
(66.2%) then constipation (43.66%). The latter was more frequent in MSA-P than MSA-C
(50.0% *vs*. 26.31%) with marginal association p = 0.06. Neurophysiological autonomic testing
revealed dysautonomia among 87.5% of all explored MSA patients mostly parasympathetic in
80.9% then sympathetic in 47.6% (S1 Table).

Upon examination, cerebellar syndrome was found in 39.44% and parkinsonism in 88.73%
mostly of PIGD phenotype in 59.15% of cases, with a median of MDS-UPDRS-III score of
29 [12–48]. Patients with MSA-P had a significantly higher frequency of dystonia and myoclo-
nus compared to MSA-C (respectively p = 0.007 and p = 0.044). Conversely, pyramidal
signs, found in 45.07% of all MSA patients, were significantly more common in MSA-C forms
(73.68% *vs*. 34.61%; p = 0.003).

Concerning imaging features, we noted significantly higher frequency of hot cross bun sign
(HCB) in patients with MSA-P compared to MSA-C forms (63.46% *vs*. 52.63%, p<0.001).

### 2. APOE genotype in MSA patients

*APOE* genotypes had different frequencies across MSA subtypes. In fact, ε3ε3 was the most
frequent (66.7%) in total MSA population as well as in both motor subtypes, followed by
ε3ε4 (21.56%). The latter was more common among MSA-P compared to MSA-C (24.4% *vs*.

**Table 1. Demographic, clinical and imaging data in MSA patients and motor subtypes.**

| Explanatory variables | Total | MSA-P | MSA-C | P value |
|---|---|---|---|---|
| | N = 71 (%) | N = 52 (%) | N = 19 (%) | |
| *Demographic variables* | | | | |
| Gender (sex-ratio) | 28/43 (0.65) | 20/32 (0.62) | 8/11 (0.72) | 0.784 |
| Age of onset | 59.11±9.01 | 59.56±9.90 | 57.89±6.02 | 0.495 |
| Age of parkinsonism onset | 62[50.5–66]* | 60.4±9.7 | 60.4±3.4 | 0.165 |
| | | 63 [55.7–67]* | 61 [58–62.5]* | |
| | | [37–77]** | [55–66]** | |
| Age of diagnosis | 62.56±8.92 | 63.27 ±9.65 | 61.0 [58–64]* | 0.273 |
| Disease duration | 3[1.0–5.0]* | 3.0[2.0–5.0]* | 2.00 [1–3]* | 0.142 |
| Parkinsonism duration | 2[1.0–4.5]* | 3.0[2.0–5.0]* | 2.0[1.25–4.0]* | **0.0051** |
| *Family History of Neurodegenerative Diseases* | | | | |
| Dementia | 16 (22.54) | 13 (25.0) | 3 (15.7) | 0.418 |
| Parkinsonism | 14 (19.72) | 12 (23.07) | 2 (10.5) | 0.245 |
| Psychiatric | 9 (12.68) | 8 (15.38) | 1 (5.26) | 0.263 |
| *Clinical data* | | | | |
| Age of ataxia *** | 54.7±8.9 | 48.7±11.1 | 57.8±5.8 | **0.009** |
| Duration of ataxia | 3.07±2.02 | 3.89±2.6 | 2.89±1.64 | 0.511 |
| | 2.0 [2–4.5]* | 2 [2–5]* | 2 [1–6]* | |
| Age of onset dysautonomic symptoms | 58 [51–65.5]* | 60.5[50.5–67]* | 58[54.5–60.5]* | 0.813 |
| Duration of dysautonomic symptoms | 2[1.0–4.5]* | 2.0[1.0–5.0]* | 2.00[1–3]* | 0.154 |
| RBD | 30 (42.25) | 21 (40.38) | 9 (47.36) | 0.604 |
| Hallucinations | 6 (8.45) | 6 (11.53) | 0 (0.0) | 0.125 |
| Frequent falls | 26 (36.62) | 17 (32.69) | 9 (47.36) | 0.262 |
| Stridor | 23 (32.39) | 17 (32.69) | 6 (31.57) | 0.930 |
| Urinary dysfunction | 61 (85.91) | 44 (84.61) | 17 (89.47) | 0.726 |
| Erectile dysfunction | 18 (25.35) | 13 (25.0) | 5 (26.31) | 0.871 |
| Constipation | 31 (43.66) | 26 (50.0) | 5 (26.31) | 0.06 |
| Orthostatic hypotension | 47 (66.2) | 34 (65.38) | 13 (68.42) | 0.891 |
| Sudomotor dysfunction | 15 (21.13) | 12 (23.07) | 3 (15.78) | 0.489 |
| *Neurological examination* | | | | |
| Parkinson syndrome | 63 (88.73) | 52 (100.0) | 11 (57.89) | **<0.001** |
| PIGD form | 42 (59.15) | 35 (67.3) | 7 (36.84) | Na |
| *Tremor dominant form* | 4 (5.63) | 4 (7.69) | 0 (0.0) | |
| *Intermediate* | 7 (9.86) | 7 (13.46) | 0 (0.0) | |
| PIGD score | 6.0[1.0–12.5]* | 8.50 [4–17]* | 2.0[1.0–7.0]* | **<0.001** |
| MDS-UPDRS-III score | 29.0 [12–48]* | 38.5 [22.2–53]* | 6.0 [5–18.0]* | **<0.001** |
| Levodopa response | 37 (52.11) | 33 (63.46) | 4 (21.05) | **0.0006** |
| Hoehn et Yahr score | 2.28±1.23 | 2.73±0.87 | 0.0 [0–2]* | **<0.001** |
| *Other movement disorders* | | | | |
| Dystonia | 20 (28.17) | 19 (36.53) | 1 (5.26) | **0.007** |
| Myoclonic jerk | 15 (21.13) | 14 (26.92) | 1 (5.26) | **0.044** |
| Cerebellar syndrome | 28 (39.44) | 9 (17.30) | 19 (100.0) | **<0.001** |
| Pyramidal signs | 32 (45.07) | 18 (34.61) | 14 (73.68) | **0.003** |
| Bulbar signs | 11 (15.49) | 6 (11.53) | 5 (26.31) | 0.08 |
| Oculomotor signs | 12 (16.90) | 9 (17.30) | 3 (15.78) | 0.856 |
| *Imaging features* | | | | |
| Cerebral atrophy | 42 (59.15) | 30 (57.69) | 12 (63.15) | 0.688 |

(*Continued*)

**Table 1.** (Continued)

| Explanatory variables | Total | MSA-P | MSA-C | P value |
|---|---|---|---|---|
| | N = 71 (%) | N = 52 (%) | N = 19 (%) | |
| Cerebellar atrophy | 17 (23.94) | 14 (26.92) | 3 (15.78) | 0.231 |
| Brainstem atrophy | 14 (19.72) | 12 (23.07) | 2 (10.52) | 0.198 |
| Hot cross bun sign | 43 (60.56) | 33 (63.46) | 10 (52.63) | **<0.001** |

*: median [1st quartile-3rd quartile]

P value[1]: value according to E4 carriage

**: [min-max]

***: the mean of age of ataxia was calculated among only 8 patients with AMS-P

MDS-UPDRS-III; MDS-Unified Parkinson's disease Rating Scale section III

10.0%). *APOE* ε3ε2 was present in 9.8% of MSA cases and finally, 1.96% of cases were carriers of ε4ε4 genotype. In summary, 23.52% of total MSA patients were carriers of at least one *APOE* -ε4 allele (24.4% of MSA-P and 20% of MSA-C) (Table 2).

## 3. Cognitive profile of MSA patients

Detailed neuropsychological characteristics of the total study population and across motor subtypes are summarized in Table 3 and illustrated in Figs 1 and 2.

CI was found in 85.7% (MCI in 73% of patients and dementia in 12.7%), and 14.3% had no cognitive impairment (NCI). In MCI patients, 37.3% had multiple-domain amnestic MCI, 29.4% single-domain non-amnestic MCI, 25.5% single-domain amnestic MCI and finally 7.8% multiple-domain non amnestic MCI. Mean initial MMSE score was 24.33±5.0 among total MSA patients and was significantly lower among MSA-P compared to MSA-C (23.52 *vs.* 26.47; p = 0.027) (S2 Table).

On neuropsychological assessment, the main altered domains were attention (64.79%) and executive functions (60.56%) which were significantly more affected in MSA-P patients compared to MSA-C (69.23% *vs.* 36.84%; p = 0.029). Mean initial FAB score was 12.31+4.88 with marginal differences between both MSA motor subtypes (p = 0.074), but only deficits in conflicting instructions subset were significantly more pronounced among MSA-P compared to

**Table 2. Comparison of *APOE* variants in MSA patients with and without dementia.**

| Genotype | MSA-P N = 41 | MSA-C N = 10 | P value | OR CI95% | MSA with MCI N = 33 | MSA with dementia N = 07 | MSA without cognitive impairment N = 10 | P[1] value | OR[1] CI95% |
|---|---|---|---|---|---|---|---|---|---|
| E4 | 10 (24.4) | 02 (20.0) | 0.07 | 1.25 (0.63–1.45) | 08 (24.3) | 01 (14.3) | 03 (30.0) | **0.04** | **1.32 (1.24–1.81)** |
| Non-E4 | 31 (75.6) | 08 (80.0) | | | 25 (75.7) | 06 (85.7) | 07 (70.0) | | |
| E3/E3 | 28 (68.3) | 06 (60.0) | | | 23 (69.7) | 04 (57.1) | 06 (60.0) | | |
| E3/E4 | 10 (24.4) | 01 (10.0) | | | 07 (21.2) | 01 (14.3) | 03 (30.0) | | |
| E4/E4 | 00 (0.0) | 01 (10.0) | | | 01 (3.03) | 00 (0.0) | 00 (0.0) | | |
| E3/E2 | 03 (7.32) | 02 (20.0) | | | 02 (6.06) | 02 (28.6) | 01 (10.0) | | |

MSA-P: Multiple system atrophy parkinsonian subtype; MSA-C: Multiple system atrophy cerebellar subtype

P value: comparison between MSA-P groups *vs.* MSA-C.

P[1] value: comparison between MSA with dementia *vs.* MSA without cognitive impairment (*APOE* ε4 carriers *vs.* non-carriers)

OR[1]: odds ratio of P[1]

CI: confidence interval in 95%

**Table 3. Cognitive profile in patients with MSA and motor subtypes.**

| Explanatory variables | Total MSA N = 71* (%) | MSA-P N = 52(%) | MSA-C N = 19(%) | P value | P value [1] |
|---|---|---|---|---|---|
| *Altered cognitive domain* | | | | | |
| Attention | 46 (64.79) | 36 (69.23) | 10 (52.63) | 0.452 | 0.685 |
| Memory | 38 (53.52) | 26 (50.0) | 12 (63.16) | 0.694 | 0.177 |
| Hippocampal profile | 12 (16.9) | 10 (19.23) | 13 (68.42) | 0.159 | 0.135 |
| Executive functions | 43 (60.56) | 36 (69.23) | 7 (36.84) | **0.029** | **0.0104** |
| Linguistic | 14 (19.72) | 12 (23.08) | 2 (10.53) | 0.147 | **0.0157** |
| Apraxia | 19 (26.76) | 16 (30.77) | 3 (15.79) | 0.290 | 0.567 |
| Visuo-spatial | 14 (19.72) | 10 (19.23) | 4 (21.05) | 0.673 | 0.991 |
| *Global cognitive status* | | | | | |
| *MMSE* | 24.33±5 | 23.52±5.51 | 26.47±2.29 | **0.027** | 0.07 |
| *Memory function: Grober & Buschke* | | | | | |
| Immediate Recall (/48) | 24.02±11.14 | 22.16±10.18 | 28.46±12.47 | 0.087 | **0.0073** |
| Delayed Recall (/16) | 6.81±3.58 | 5.97±3.63 | 8.69±2.75 | **0.020** | **0.05** |
| Word list saving (%) | 78.01 | 74.32 | 87.21 | 0.104 | 0.0814 |
| *Executive function: FAB* | 12.31±4.88 | 11.67±4.73 | 14.12±4.97 | 0.074 | **0.046** |
| Similarities | 26 (36.62) | 21 (40.38) | 5 (26.32) | 0.322 | 0.102 |
| Similarities Score | 2.39±0.89 | 2.37±0.88 | 2.44±0.96 | 0.795 | 0.770 |
| Lexical Fluency | 28 (39.44) | 23 (44.23) | 5 (26.32) | 0.200 | 0.311 |
| Lexical Fluency Score | 2.03±1.20 | 1.91±1.24 | 2.38±1.02 | 0.187 | 0.348 |
| Motor series « Luria » test | 27 (38.03) | 21 (40.38) | 6 (31.58) | 0.578 | 0.230 |
| Motor series « Luria » Score | 2.05±1.17 | 2.02±1.16 | 2.12±1.2 | 0.762 | 0.534 |
| Conflicting instruction | 32 (45.07) | 27 (51.92) | 5 (26.32) | **0.048** | **0.011** |
| Conflicting instruction Score | 2.0 [0.0–3.0]* | 1.0[0.0–3.0]* | 2.31±1.2 | **0.031** | **0.042** |
| Go-No Go | 35 (49.30) | 28 (53.85) | 7 (36.84) | 0.205 | 0.274 |
| Go-No Go Score | 1.48±1.40 | 1.24±1.4 | 2.12±1.2 | **0.029** | 0.063 |
| Prehension behaviour Score | 2.9±0.53 | 2.87±0.62 | 3.0±0.0 | 0.404 | 0.580 |
| *Mood and behavioural* | | | | | |
| Mood disorders | 51 (71.83) | 40 (76.92) | 11 (57.89) | **0.035** | 0.252 |
| GDS score | 13.72±6.13 | 13.52±5.46 | 14.75±10.05 | 0.722 | 0.318 |

*: median [1st quartile-3rd quartile]

P value[1]: p value according to E4 carriage

MSA-C (51.92 *vs.* 26.32; p = 0.048). Similarly, mood disorders were more frequently associated to MSA-P (p = 0.035).

## 3. Determinants of cognitive impairment in MSA patients

**3.1. Clinical and paraclinical determinants of cognitive impairment in MSA.** Disease duration was significantly different across NCI, MCI and dementia subgroups (p = 0.005), with longer disease duration among demented MSA. The prevalence of PIGD phenotype of parkinsonism was higher among patients with dementia (88.9%) compared to NCI and MCI subgroups (60% and 54.9%). Indeed, significantly higher PIGD and MDS-UPDRS-III scores were noted in demented MSA patients (p = 0.019 and p = 0.015 respectively). The clinical and neurophysiological study of the autonomic system revealed no differences across the cognitive subtypes.

The evaluation of the correlation between cognitive scores and clinical/neurophysiological data revealed that MMSE and FAB initial scores were negatively correlated with MDS-UPDRS-III

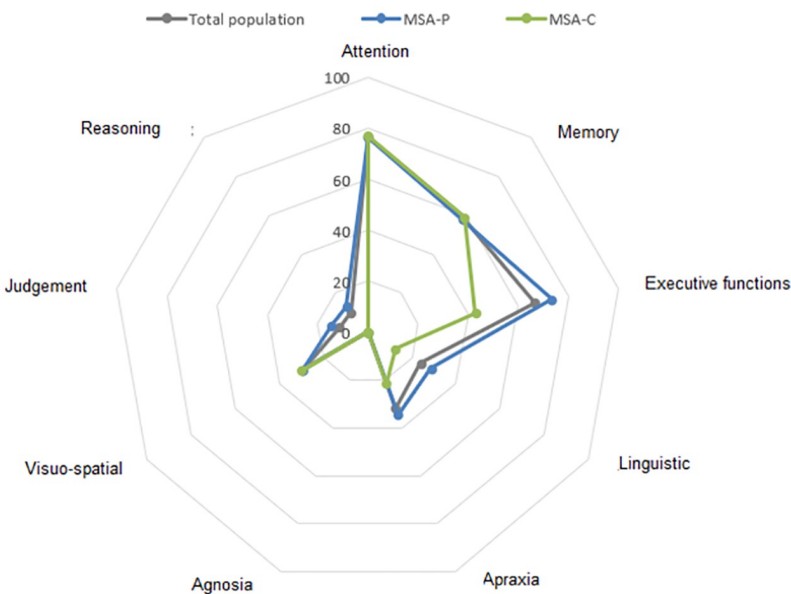

**Fig 1. Comparison of altered cognitive domains in multiple system atrophy according to motor subtype (MSA-P vs MSA-C and total study cohort).** Radar charts comparing the percentage (%) of AMS-C and AMS-P cases with neurophysiological findings.

score in all MSA patients and in both motor subtypes. Both scores were inversely proportional to ataxia duration in the total cohort and in MSA-P. Moreover, MMSE and FAB were also inversely proportional to disease and dysautonomia duration in only MSA-C patients. A negative correlation was also noted between MMSE score and parkinsonism duration in MSA-C patients.

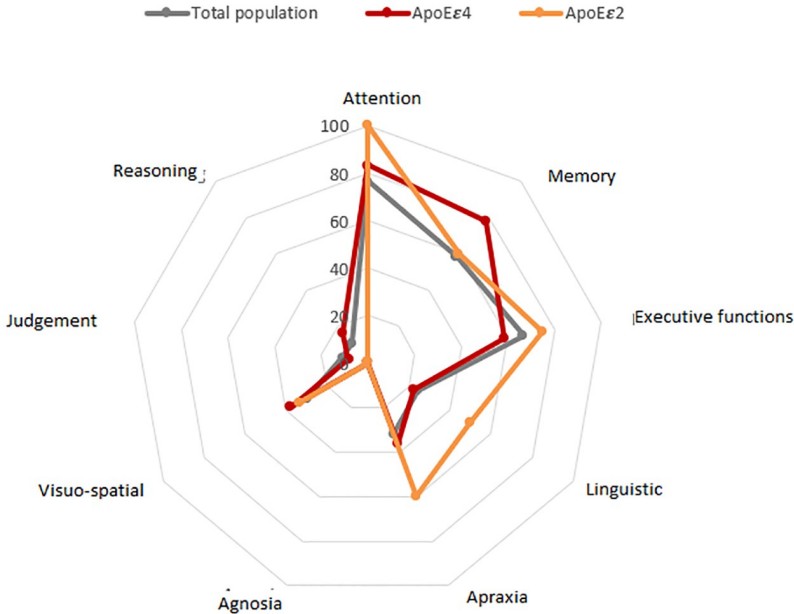

**Fig 2. Remodeling of cognitive profile according to the carriage of APOEε4 and APOEε2 isoforms.** Radar charts comparing the altered cognitive domains in percentage (%) between two subtypes of AMS-C and AMS-P.

**3.2. ApoEε4 effect on cognitive profile in MSA.** MSA- *APOE* ε4 carriers had 1.32 fold higher risk of developing CI compared to non-carriers (p = 0.04) (Table 3). By combining the *APOE* ε4 allele carriage status, NCI-MSA patients had significantly younger age of onset of the disease, of parkinsonism and dysautonomia compared to other cognitive subgroups (p = 0.05, p = 0.043 and p = 0.05 respectively).

CI severity was more pronounced in *APOE* ε4 carriers who had lower initial MMSE scores (p = 0.0001). When analyzing the cognitive profile, the presence of altered attention was statistically significant by adjusting the p value to *APOE* ε4 carriers (p = 0.046), as well as language deficits (p = 0.044). Moreover, by considering the status of carriage of *APOE* ε4 allele, significant differences appeared between MSA motor subtypes for language impairment (p = 0.0157) and FAB initial score (p = 0.046) which were more altered in MSA-P. Indeed, executive dysfunction was more pronounced in MSA-P *APOE* ε4 carriers (p = 0.0104) (Fig 2).

The evaluation of the correlation between clinical/neurophysiological data and *APOE* ε4 carrying status revealed a correlation of the carriage of *APOE* ε4 allele and ataxia duration in MSA-P patients.

## Discussion

In this study, we depicted the cognitive profile of Tunisian patients with MSA and provided first data on clinical and genetic determinants of CI. As reported in our recent epidemiological study on Tunisian APS [34], we noted a predominance of female gender and MSA-P motor subtype mostly of PIGD phenotype.

Interestingly, a levodopa responsiveness was found in 52.1% of total MSA patients, especially in MSA-P (63.5%) vs and MSA-C (21.1%). Although a poor levodopa response is one of the consensus criteria for the diagnosis of MSA and helps to differentiate MSA from PD, approximately one third of patients may experience some benefit, especially if they have MSA-P, as in our patients. Indeed, in pathologically-confirmed MSA series, a positive L-dopa response was reported in 28% to 65% of patients, persisting for several years in only 13% of cases. Hence, the rate of effectiveness of levodopa found in our series is still within this range [35]. However, the degree of responsiveness is difficult to standardize between the existing studies, as several methods and scales were used across them, going from the ALC [36, 37] to prolonged levodopa intake at different doses and durations [38–42]. In our study, for the assessment of levodopa responsiveness, we used the ALC that has been proven to predict levodopa chronic responsiveness [43].

The stratification of our MSA patients according to the presence and the severity of CI revealed the high prevalence of such disorders (85.7%) mostly of mild severity (73%), and only 12.7% had dementia. The percentages of MCI and dementia in our MSA patients were within the ranges of other studies (respectively 32.7% to 80% for MCI and 5 to 31% for dementia [4, 44–46]. However, the frequency of CI in our clinically diagnosed cohort was higher than that reported in 102 autopsy-confirmed MSA patients (32%) [7]. Differences in definitions, methods and study designs could explain such discrepancies.

Beyond the determination of CI prevalence in MSA, recent studies tended to further analyze the cognitive profile in this disease revealing the broad spectrum of encountered deficits.

Hong et al. have illustrated this cognitive heterogeneity by stratifying MCI-MSA patients into four categories. Along with their results, we found the same distribution of MCI patients with a predominance of multiple-domain amnestic MCI, followed by single-domain non-amnestic MCI, and single-domain amnestic MCI. In our cohort, 7.8% of MCI patients had multiple-domain non-amnestic MCI, though none of the patients in the study of Hong et al. developed such a profile [47].

In our MSA patients, the main altered cognitive domains were attention (64.79%), executive functions (60.56%) and to a lesser extent memory. These three domains were effectively predominantly altered in several previous studies [45, 48–50]. Indeed, executive dysfunction was described in 49–69% of MSA patients [35, 50], and was more marked in MSA-P as found in our cohort (p = 0.029).

Dysexecutive syndrome was reported to comprise in details a decline in semantic and phonemic verbal fluency [4, 47, 48], preservative behavior, problem solving, flexibility, response inhibition, attention and working memory [4, 14, 16, 51]. Yet, data related to the most affected subsets of executive functions are divergent. Santangelo et al. reported that spatial planning was most compromised, followed by cognitive flexibility [14]. On the contrary, Cao et al. found that inhibitory control was the most impaired (60.9%) which corroborate to the results of our cohort where inhibitor control dysfunction reached 49% of cases [50]. Some other studies reported no impairment in verbal or nonverbal fluency [14].

Memory was affected in 65.2% of our patients which is in agreement with previous studies where it varied from 17 to 66% [3, 45, 47]. Memory deficits concerned verbal learning, immediate and delayed recall and less often recognition [46, 49].

With regards to visuospatial functions, previous studies reported conflicting results, with some suggesting preserved function [51] while others found deficits in this domain [4, 48, 49]. Visuospatial dysfunction was noted in 34.5% of our patients almost within the range previously described (10–30%) [44, 45]. Language dysfunction and apraxia were less frequently found respectively in 10–20% [4, 45], and 8–10% [52]. Interestingly, we noted the presence of hallucinations in 8.45% of our MSA patients. Indeed, hallucinations not induced by drugs are considered as non-supporting features of MSA. However, according to previous reports, the frequency of MSA patients experiencing hallucinations ranged from 5.5% in the European Multiple System Atrophy registry, to 15% in studies where Neuropsychiatric Inventory was used as in our study [53].

The predictive factors of CI are underexplored. In our study, MSA patients with dementia had longer disease and parkinsonism duration compared to patients without dementia. Disease duration was only related to executive dysfunction. Moreover, decline of global cognitive function and executive function were correlated to duration of parkinsonism and ataxia as well as motor disability. These factors have been previously reported in the literature. In fact, Hatakeyama et al. demonstrated that impairment of global cognitive function (MMSE score) was related to a long disease duration [17]. Furthermore, it has been reported that among MSA patients surviving more than 8 years, almost 50% had CI and that 14% of MSA patients were demented in the last year prior to death [52]. Nevertheless, age at onset, like in our cohort, did not seem to influence cognition [46]. According to Kitayama et al., there were no significant differences in age at onset, gender, duration of disease, or severity of cerebellar dysfunction between patients with and without dementia [20]. Regarding the other demographic features, cognitive and behavioral differences have been reported to be more compromised in MSA females [54]. Motor disability is also a major factor correlated with CI and dementia in MSA [47, 46]. Effectively, the severity of motor disability was associated with MMSE [46, 44], and FAB scores [17] as it is in our study. Otherwise, data regarding the influence of MSA motor subtype are sparse. Indeed, some observed more pronounced cognitive dysfunction in MSA-P [48], and conversely other studies showed a more prominent decline of executive function and verbal memory in MSA-C [14, 55, 56], while a third group found comparable profile in both subtypes [49]. As regards autonomic functions, we did not find any correlations with CI in our series either clinically or neurophysiologically. Few studies addressed orthostatic hypotension with controversial findings [16]. However, cardiovascular dysautonomia have been reported [12],

and MMSE was positively correlated to heart rate variation on neurophysiological assessment [17].

The genetic screening of MSA for *APOE* ε4 revealed its association to CI occurrence (p = 0.04) and severity with lower MMSE score (p = 0.0001). *APOE* is well established lipid metabolism gene and the major genetic determinant of late onset AD [21]. *APOE* has also been significantly correlated to reduced α-synuclein uptake in an isoforms-dependant manner through direct regulation of α-synuclein pathology independently of its established effects on Aβ and tau [21]. Furthermore, MSA is characterized by the misfolded α-synuclein leading to formation of GCIs [57]. In addition, numerous GCIs were found in the underlying white matter in MSA patients with CI [58]. Armstrong et al. found vacuolation of GCIs in the frontal cortex in MSA patients with CI [59]. Taken all together, these findings prompt the speculations to the association of *APOE* ε4 isoform with the disruption of GCIs, the pathological hallmark of MSA, via α-synuclein misfolding, leading to CI among MSA patients.

Mostly impaired cognitive domains in MSA patients were attention (p = 0.046) and language (p = 0.044) with *APOE* adjustment. Moreover, executive dysfunction was more pronounced among MSA-P *APOE* ε4 carriers (p = 0.010). Carrying *APOE* ε4 allele was previously reported to be correlated with the presence of executive dysfunction in PD patients [21], which may explain the more marked executive deficits in our MSA-P patients. Moreover, imaging studies on cognitively impaired MSA patients, showed a significant atrophy in the left dorsolateral prefrontal cortex [7, 44]. The potential mechanism underlying the executive impairment is the lower prefrontal volume and the decreased neuronal activity in prefrontal areas in *APOE* ε4 carriers [60].

We also reported in our study the significant correlation of ataxia duration with the carriage of *APOE ε4* isoform. These findings could be explained by the regionally specific *APOE* mRNA transcription in neurons. Indeed, in human cerebellar cortex, very strong *APOE* mRNA hybridization signal was observed in Bergmann radial glial cells [61], and a strong expression of apolipoprotein E receptor 2 in Purkinje cells of the cerebellum [62]. All these cells and structures are implicated in the occurrence of ataxia possibly modulated by *APOE*.

There are number of flaws in the current work that should be addressed. The small size of our sample and the inequality between both MSA motor subtypes may reduce the statistical power of our study. Similarly, the lack of *APOE* genotyping in all included patients and the lack of neuropathological confirmation may be major limitations of this study. In fact, our cohort comprised only clinically-diagnosed MSA patients exposing to the risk of diagnostic uncertainty because of heterogeneous clinical presentations of MSA and possible presence of MSA-look-alikes. Moreover, the new MDS criteria for the diagnosis of MSA of 2022 could not be verified in our cohort, since the cases included in the study may not be rechecked for the new requirements mainly for those of orthostatic hypotension and urinary dysfunction [63].

## Conclusion

This is the first Tunisian study conducted to investigate the presence of CI in MSA and the prevalence of dementia in MSA in a Tunisian cohort. In our cohort, MSA-P, mainly of PIGD phenotype, disease duration and *APOE* ε4 carrying status appeared as the main determinants of CI in Tunisian MSA patients, defining a more altered cognitive phenotype. This effect mainly concerned executive, attention and language functions, all found to be more impaired in *APOE* ε4 carriers with variable degrees across MSA motor subtypes. Dysautonomia did not seem to play a paramount role in the occurrence nor in the profile of CI in our MSA patients. Further studies on larger samples are mandatory to analyze more extensively predictive genetic

factors of cognitive decline. Prospective follow-up is also needed to assess the evolving profile of such deficits and their prognostic determinants.

## Supporting information

**S1 Table. Comparison of SSR and Neurophysiological cardiovascular autonomic testing in MSA patients with and without dementia.**
(DOCX)

**S2 Table. Comparison of demographic, clinical and cerebral imaging data in MSA patients according to cognitive impairment level (without, mild and major).**
(DOCX)

## Acknowledgments

We thank all blood donor patients, who consented and participated in the present study. We would like to express our gratitude to the sequencing Platform in Faculty of Medicine of Tunis (FMT) for their contribution and their excellent assistance in experiments.

## Author Contributions

**Conceptualization:** Amina Nasri, Amina Gargouri, Mouna Ben Djebara, Imen Kacem, Riadh Gouider.

**Data curation:** Amina Nasri, Alya Gharbi, Ikram Sghaier, Saloua Mrabet, Amira Souissi.

**Formal analysis:** Amina Nasri, Alya Gharbi, Ikram Sghaier, Saloua Mrabet, Amira Souissi.

**Investigation:** Amina Nasri.

**Methodology:** Amina Nasri, Ikram Sghaier, Riadh Gouider.

**Project administration:** Amina Nasri, Riadh Gouider.

**Supervision:** Mouna Ben Djebara, Imen Kacem, Riadh Gouider.

**Validation:** Amina Gargouri, Mouna Ben Djebara, Imen Kacem, Riadh Gouider.

**Visualization:** Riadh Gouider.

**Writing – original draft:** Amina Nasri, Alya Gharbi, Ikram Sghaier.

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
