## [Decision Letter · Decision Letter 0]

30 Mar 2022

PONE-D-21-32170Determinants of cognitive impairment in Multiple System Atrophy: Clinical and genetic studyPLOS ONE

Dear Dr. Gouider,

Thank you for submitting your manuscript to PLOS ONE. After careful consideration, we feel that it has merit but does not fully meet PLOS ONE’s publication criteria as it currently stands. Therefore, we invite you to submit a revised version of the manuscript that addresses the points raised during the review process.

We look forward to receiving your revised manuscript.

Kind regards,

Tuhin Virmani

Academic Editor

PLOS ONE

Journal Requirements:

Additional Editor Comments:

Dear Dr. Gouider,

I apologize for the delay in your manuscript but it has been difficult to secure reviewers. I have therefore decided to proceed with the reviews of the single reviewer. If you decide to resubmit based on those requests for major revision, your manuscript will need to be sent to this reviewer and possibly additional reviewers if they can be obtained.

Best regards,

Tuhin Virmani

Reviewers' comments:

Reviewer's Responses to Questions

**Comments to the Author**

1. Is the manuscript technically sound, and do the data support the conclusions?

Reviewer #1: No

2. Has the statistical analysis been performed appropriately and rigorously? 

Reviewer #1: No

3. Have the authors made all data underlying the findings in their manuscript fully available?

Reviewer #1: No

4. Is the manuscript presented in an intelligible fashion and written in standard English?

Reviewer #1: Yes

5. Review Comments to the Author

Reviewer #1: This manuscript appears to have been submitted in the form of a very early draft. Multiple problematic tables are presented, some of which are larger than a page and difficult to read. There are numerical errors and cells with omitted data. In some places means are provided, in some places medians, in some places both. Means are more appropriate. There are problems with the statistical analysis. The figures lack proper explanation. There are major issues with the study design. What proportion of subjects had possible vs probable MSA by the 2008 criteria? (highly relevant to diagnostic accuracy.) How long were these patients followed? Was the diagnosis made at the first or the last visit? What is the explanation for the high percentage of hallucinations? High percentage of levodopa response in some groups? I would recommend that the authors make major revisions to all sections of the manuscript, and enlist the expertise of a statistician.

6. PLOS authors have the option to publish the peer review history of their article (what does this mean?). If published, this will include your full peer review and any attached files.

Reviewer #1: No

---

## [Author Response · Author response to Decision Letter 0]

4 Jun 2022

EDITORS’ COMMENTS

EDITOR: 

Editor: Thank you for submitting your manuscript to PLOS ONE. After careful consideration, we feel that it has merit but does not fully meet PLOS ONE’s publication criteria as it currently stands. Therefore, we invite you to submit a revised version of the manuscript that addresses the points raised during the review process. 

Authors: We appreciate the editor’s response to our submission and we believe that the modifications and revisions conducted in the manuscript will enhance the paper quality and we agree with all the comments raised by the reviewer. We hope that our revision will be suitable for publication in PLOS ONE journal. 

RESPONSE TO REVIEWERS’ COMMENTS

REVIEWER #1: 

Reviewer: This manuscript appears to have been submitted in the form of a very early draft. Multiple problematic tables are presented, some of which are larger than a page and difficult to read. There are numerical errors and cells with omitted data.

Authors: Done, we agree with the reviewer comments and we revised the tables. Numerical errors were addressed and the empty cells were filled. We adapted the tables to the page dimensions and we hope that our revision made the data clearer and our modifications will satisfy the reviewer. 

Reviewer: In some places means are provided, in some places medians, in some places both. Means are more appropriate. There are problems with the statistical analysis.

Authors: Indeed, we generally used the means and +/- the standard deviation (SD). However, we purposely used the median in some cases because the data was not normally distributed. Hence, from a statistically point of view the median and the 1st quartile-3rd quartile are more appropriate. We added this explanation in the material and methods part (section statistics) and we discussed the issue with professional statistician who confirmed our methodology.

Reviewer: The figures lack proper explanation.

Authors: Done, we added further detailed explanation to the figure as suggested by the reviewer

Reviewer: There are major issues with the study design. What proportion of subjects had possible vs probable MSA by the 2008 criteria? (highly relevant to diagnostic accuracy.)

Authors: Done, we added the proportions of subjects with possible vs probable MSA by the 2008 criteria and specified the dysautonomic features that allowed this probability subtyping in the manuscript in the part of the methods and the results’parts. We are thankful for the reviewer’s careful revision.

Reviewer: How long were these patients followed? Was the diagnosis made at the first or the last visit?

Authors: Done, we added the mean follow-up of our cohort. It is worthy to note that the follow-up duration was calculated as the difference between the age at the last visit (or the age of death) and the age at the first consultation to our department. Furthermore, diagnostic delay was defined as the difference between the age at first consultation and the age of disease onset. These details were added to the methods ‘part. The diagnosis of MSA was raised mostly at first consultation and finally retained after performing the required work-up in order to discard other differential diagnosis or after a period of follow-up/ levodopa treatment showing subsequent resistance to levodopa or the emergence of additional clinical features favoring MSA diagnosis. The mean delay between first consultation and MSA diagnosis in our cohort was 1.42 years ranging from 0 (diagnosis made at first consultation) to 5 years. 

Reviewer: What is the explanation for the high percentage of hallucinations?

Authors: We thank the reviewer for this interesting point that has been raised. Indeed, hallucinations not induced by drugs are considered as non-supporting features of MSA. However, according to previous reports, the frequency of MSA patients experiencing hallucinations ranged from 5.5% in the European Multiple System Atrophy registry, to 15% in studies where Neuropsychiatric Inventory was used as in our study [Watanabe H, et al, J Mov Disord. 2018]. All these details along with the corresponding reference were added to the discussion section.

Reviewer: What is the explanation High percentage of levodopa response in some groups?

Authors: Again, we are thankful for the careful review. Indeed, a poor levodopa response is one of the consensus criteria for the diagnosis of MSA and helps to differentiate MSA from Parkinson’s disease (PD). Nonetheless, approximately one third of patients may experience some benefit, especially if they have MSA-P, as in our patients. Indeed, in pathologically-confirmed MSA series, a positive L-dopa response was reported in 28% to 65% of patients, persisting for several years in only 13% of cases (Perez-Lloret et al. Mov Disord Clin Pract, 2015). Hence, the rate of effectiveness of levodopa found in our series is still within this range (Total MSA patients (52.1%), MSA-P (63.5%) and MSA-C (21.1%). 

Reviewer: I would recommend that the authors make major revisions to all sections of the manuscript, and enlist the expertise of a statistician.

Authors: Done, all the sections of the manuscript were revised according to the comments made by the reviewer and the tables were revised by statistician and validated by all co-authors after discussion of the revised version of tables. We hope the changes made would satisfy the reviewer.

---

## [Decision Letter · Decision Letter 1]

31 Aug 2022

PONE-D-21-32170R1Determinants of cognitive impairment in Multiple System Atrophy: Clinical and genetic studyPLOS ONE

Dear Dr. Gouider,

Thank you for submitting your manuscript to PLOS ONE. After careful consideration, we feel that it has merit but does not fully meet PLOS ONE’s publication criteria as it currently stands. Therefore, we invite you to submit a revised version of the manuscript that addresses the points raised during the review process.

We look forward to receiving your revised manuscript.

Kind regards,

Tuhin Virmani, MD, PhD

Academic Editor

PLOS ONE

Journal Requirements:

Reviewers' comments:

Reviewer's Responses to Questions

**Comments to the Author**

1. If the authors have adequately addressed your comments raised in a previous round of review and you feel that this manuscript is now acceptable for publication, you may indicate that here to bypass the “Comments to the Author” section, enter your conflict of interest statement in the “Confidential to Editor” section, and submit your "Accept" recommendation.

Reviewer #1: (No Response)

2. Is the manuscript technically sound, and do the data support the conclusions?

Reviewer #1: Partly

3. Has the statistical analysis been performed appropriately and rigorously? 

Reviewer #1: Yes

4. Have the authors made all data underlying the findings in their manuscript fully available?

Reviewer #1: Yes

5. Is the manuscript presented in an intelligible fashion and written in standard English?

Reviewer #1: Yes

6. Review Comments to the Author

Reviewer #1: In the last year new MDS consensus criteria for the diagnosis of MSA were published.

Wenning GK et al. The Movement Disorder Society Criteria for the Diagnosis of Multiple System Atrophy. Mov Disord. 2022 Jun;37(6):1131-1148. doi: 10.1002/mds.29005. Epub 2022 Apr 21. PMID: 35445419; PMCID: PMC9321158.

What proportion of subjects had clinically established, clinically probable, or possible prodromal MSA? The manuscript should be updated to this new criteria.

Table 1:

I am confused by several sections:

History of Neurodegenerative diseases: what does this mean? If this is a personal history, one would expect 100% of the MSA-P group to have parkinsonism for example.

Age and duration of ataxia – Why is there such a high prevalence of ataxia in the MSA-P group?

All MSA-C patients had hallucinations?

UPDRS score – I assume this means the MDS-UPDRS III score? This should be specified.

How was levodopa-response defined?

Table 2: consider omitting

Table 5: consider omitting

Figure 2: this should be omitted or at minimum explained more adequately. It is difficult to know what one is looking at here

Regarding levodopa responsivity. The author’s response is poorly researched. Their statement about 28-65% response comes from a review article in Movement Disorders Clinical Practice that references 4 actual studies. Of those studies, by far the best quality is Wenning GK, Tison F, Ben Shlomo Y, Daniel SE, Quinn NP. Multiple system atrophy: a review of 203 pathologically proven cases. Mov Disord 1997;12:133–147 which reports 28% of patients responding to levodopa which is closer to the true number. A better explanation is warranted.

In the discussion you cite a study (reference 38, Cao et al) that you say reported 32% cognitive impairment in “autopsy-confirmed MSA patients.” Reference 38 was not an autopsy confirmed study! Those patients were diagnosed clinically and not pathologically-confirmed. This must be revised.

In the conclusion you neglect to mention disease duration as a major determinant for cognitive impairment.

7. PLOS authors have the option to publish the peer review history of their article (what does this mean?). If published, this will include your full peer review and any attached files.

Reviewer #1: No

---

## [Author Response · Author response to Decision Letter 1]

1 Oct 2022

RESPONSE TO THE EDITORS’ COMMENTS

EDITOR: 

Editor: Thank you for submitting your manuscript to PLOS ONE. After careful consideration, we feel that it has merit but does not fully meet PLOS ONE’s publication criteria as it currently stands. Therefore, we invite you to submit a revised version of the manuscript that addresses the points raised during the review process.

Authors: We appreciate the editor’s response to our submission and we believe that the modifications and revision conducted in the manuscript will enhance the paper quality and we agree with all the comments given by the reviewer. We hope that our revision will be suitable for publication in PLOS ONE journal. 

RESPONSE TO REVIEWERS’ COMMENTS

REVIEWER #1: 

Reviewer: In the last year new MDS consensus criteria for the diagnosis of MSA were published.

Wenning GK et al. The Movement Disorder Society Criteria for the Diagnosis of Multiple System Atrophy. Mov Disord. 2022 Jun;37(6):1131-1148. doi: 10.1002/mds.29005. Epub 2022 Apr 21. PMID: 35445419; PMCID: PMC9321158.

What proportion of subjects had clinically established, clinically probable, or possible prodromal MSA? The manuscript should be updated to this new criteria.

Authors: We appreciate the reviewer suggestion. Indeed, the new MDS consensus criteria for the diagnosis of MSA were published in 2022, while our study was conducted between 2003 and 2020. Therefore, we applied the Gilman criteria of 2008 (doi: 10.1212/01.wnl.0000324625.00404.15). Accordingly, our patients were stratified into 

1- probable (for autonomic dysfunction: inability to control the release of urine from the bladder, with erectile dysfunction in males) or an orthostatic decrease of blood pressure within 3 min of standing by at least 30 mmHg systolic or15 mmHg diastolic) and

2- possible MSA (for autonomic dysfunction: otherwise unexplained urinary urgency, frequency or incomplete bladder emptying, erectile dysfunction in males, or significant orthostatic blood pressure decline that does not meet the level required in probable MSA). 

However, according to the New Criteria (2022), the stratification of MSA (clinically established /clinically probable) cannot be verified in our cohort, since the cases included in the study may not be rechecked for the requirements of the new criteria, mainly for those of orthostatic hypotension and urinary dysfunction.

1-clinically established: for instance, Neurogenic OH (≥20/10 mmHg blood pressure drop) within3 minutes of standing or head-up tilt test), Unexplained voiding difficulties with post-void urinary residual volume ≥100 mL) 

2- clinically probable: Neurogenic OH (≥20/10 mmHg blood pressure drop) within 10 minutes of standing or head-up tilt test). 

Nonetheless, we mentioned this point in the limitation part of our manuscript. 

Reviewer: Table 1:

I am confused by several sections:

- History of Neurodegenerative diseases: what does this mean? If this is a personal history, one would expect 100% of the MSA-P group to have parkinsonism for example.

Authors: We thank the reviewer for the careful revision. We meant by history of neurodegenerative diseases, the family history of neurodegenerative diseases. We added this information in the table 1. 

Reviewer: Table 1:

 - Age and duration of ataxia – Why is there such a high prevalence of ataxia in the MSA-P group?

Authors: In the study of Roncevic et al (doi:10.1007/s00702-013-1133-7.), where included 100 MSA patients were included and stratified into 60 MSA-P and 40 MSA-C (Four patients eventually had post mortem neuropathological confirmation), the authors reported 32% out of the 60 MSA-P patients had Gait Ataxia and 25% of MSA-P exhibit a limb ataxia. These frequencies are higher than our findings (15% (8 out of 52) of our MSA-P patients).

Reviewer: Table 1:

 - All MSA-C patients had hallucinations?

Authors: We appreciate the reviewer’s careful revision. Indeed, the totality of MSA-C cases did not have hallucinations, and it was a typos mistake as the frequencies are correct for the total population of MSA ((6 out of 71) 8.45%). We corrected the typos in the table, which has been added in the supplementary materials according to the reviewer’s instructions.

Reviewer: Table 1:

-UPDRS score – I assume this means the MDS-UPDRS III score? This should be specified.

Authors: Done, we are grateful for the detailed revision of the reviewer. We added this specification in the methods’ part as well as in the table footnote. 

Reviewer: Table 1:

 -How was levodopa-response defined?

Authors: Done. We defined the levodopa-response in the methods section. 

Reviewer: Table 2: consider omitting

 Table 5: consider omitting

Authors: Done the tables were omitted from the manuscript and submitted as supplementary data.

Reviewer: Figure 2: this should be omitted or at minimum explained more adequately. It is difficult to know what one is looking at here

Authors: Done, we omitted the figure.

Reviewer: Regarding levodopa responsivity. The author’s response is poorly researched. Their statement about 28-65% response comes from a review article in Movement Disorders Clinical Practice that references 4 actual studies. Of those studies, by far the best quality is Wenning GK, Tison F, Ben Shlomo Y, Daniel SE, Quinn NP. Multiple system atrophy: a review of 203 pathologically proven cases. Mov Disord 1997;12:133–147 which reports 28% of patients responding to levodopa which is closer to the true number. A better explanation is warranted.

Authors: We totally agree with the reviewer. However, we would like to mention that the degree of responsiveness is difficult to standardize between the existing studies. In fact, for the study of Wenning et al., 1997 which is a review and a collection of other reports of MSA cases pathologically proven, the responsiveness was considered “where available” as the best and latest response to levodopa without mentioning the details of such classification. Moreover, in the same study 28% of treated patients showed a good or excellent response to levodopa at some stage of their treatment, but for our current study, we assessed the initial levodopa response of MSA cases. Hence the discrepancy would be expected between our study and Wenning et al review. Furthermore, several methods and scales were used in the literature, going from acute levodopa challenge (ALC) (doi:10.1097/00002826-199304000-00006 and doi:10.1136/jnnp.55.11.1009) to prolonged levodopa intake at different doses and durations (doi:10.1001/archneur.1995.00540270090024, doi:10.1093/brain/113.6.1823, doi:10.1097/00002826-199012000-00007, doi:10.1002/mds.870120203, and doi:10.1093/brain/117.4.835). For the current study, we used the ALC at the dose of 250mg of levodopa to assess levodopa responsiveness. Indeed, it has been established that acute levodopa challenge may be performed to predict levodopa chronic responsiveness (doi:10.1371/journal.pone.0172145). We discussed further our finding in the discussion part and we hope that our revision will satisfy the reviewer.

Reviewer: In the discussion you cite a study (reference 38, Cao et al) that you say reported 32% cognitive impairment in “autopsy-confirmed MSA patients.” Reference 38 was not an autopsy confirmed study! Those patients were diagnosed clinically and not pathologically-confirmed. This must be revised.

Authors: We thank the reviewer for the comment. Indeed, the references did not correspond to the sentence. We changed the manuscript accordingly.

Reviewer: In the conclusion, you neglect to mention disease duration as a major determinant for cognitive impairment.

Authors: Done, we modified the sentence accordingly. 

We kindly thank the reviewer and the editor for their comments, which we believe have greatly contributed to enriching the content of our paper and clarifying some confusing statements.

---

## [Decision Letter · Decision Letter 2]

2 Nov 2022

PONE-D-21-32170R2Determinants of cognitive impairment in Multiple System Atrophy: Clinical and genetic studyPLOS ONE

Dear Dr. Gouider,

Thank you for submitting your manuscript to PLOS ONE. After careful consideration, we feel that it has merit but does not fully meet PLOS ONE’s publication criteria as it currently stands. Therefore, we invite you to submit a revised version of the manuscript that addresses the points raised during the review process.

 As per the reviewer, the authors had reported that revisions had been made to Table 1, however the submitted version did not reflect those changes. Please make the changes as reported.

We look forward to receiving your revised manuscript.

Kind regards,

Tuhin Virmani

Academic Editor

PLOS ONE

Journal Requirements:

Reviewers' comments:

Reviewer's Responses to Questions

**Comments to the Author**

1. If the authors have adequately addressed your comments raised in a previous round of review and you feel that this manuscript is now acceptable for publication, you may indicate that here to bypass the “Comments to the Author” section, enter your conflict of interest statement in the “Confidential to Editor” section, and submit your "Accept" recommendation.

Reviewer #1: (No Response)

2. Is the manuscript technically sound, and do the data support the conclusions?

Reviewer #1: Yes

3. Has the statistical analysis been performed appropriately and rigorously? 

Reviewer #1: Yes

4. Have the authors made all data underlying the findings in their manuscript fully available?

Reviewer #1: Yes

5. Is the manuscript presented in an intelligible fashion and written in standard English?

Reviewer #1: Yes

6. Review Comments to the Author

Reviewer #1: All comments were addressed with one exception:

Table 1: The authors responded to a question about the "History of Neurodegenerative Diseases" section by saying that they meant "Family History of Neurodegenerative Diseases." In their response the authors said they had made the revision to Table 1, but it does not appear that this revision was made.

7. PLOS authors have the option to publish the peer review history of their article (what does this mean?). If published, this will include your full peer review and any attached files.

Reviewer #1: No

---

## [Author Response · Author response to Decision Letter 2]

3 Nov 2022

RESPONSE TO THE EDITORS’ COMMENTS

EDITOR: 

Editor: Thank you for submitting your manuscript to PLOS ONE. After careful consideration, we feel that it has merit but does not fully meet PLOS ONE’s publication criteria as it currently stands. Therefore, we invite you to submit a revised version of the manuscript that addresses the points raised during the review process. As per the reviewer, the authors had reported that revisions had been made to Table 1, however the submitted version did not reflect those changes. Please make the changes as reported.

Authors: We appreciate the editor’s response to our submission. The changes requested by the reviewer have been made. We hope that our revision will be suitable for publication in PLOS ONE journal. 

RESPONSE TO REVIEWERS’ COMMENTS

REVIEWER #1: 

Reviewer: All comments were addressed with one exception: Table 1: The authors responded to a question about the "History of Neurodegenerative Diseases" section by saying that they meant "Family History of Neurodegenerative Diseases." In their response the authors said they had made the revision to Table 1, but it does not appear that this revision was made.

Authors: We thank the reviewer for the careful revision. We made the requested changes and added this information in Table 1.

. 

We kindly thank the reviewer and the editor for their comments throughout the whole revision process, which we believe have greatly contributed to enriching the content of our paper.

---

## [Editor Report · Decision Letter 3]

4 Nov 2022

Determinants of cognitive impairment in Multiple System Atrophy: Clinical and genetic study

PONE-D-21-32170R3

Dear Dr. Gouider,

We’re pleased to inform you that your manuscript has been judged scientifically suitable for publication and will be formally accepted for publication once it meets all outstanding technical requirements.

Kind regards,

Tuhin Virmani

Academic Editor

PLOS ONE